# Cardiovascular risk among middle-aged Japanese adults with atopic dermatitis: A nested case–control study

Misato Maeno[1☉], Mami Ishida[iD][2☉*], Risa Tamagawa-Mineoka[1], Naoyuki Takashima[2], Hiroshi Ikai[3]

1 Department of Dermatology, Graduate School of Medical Sciences, Kyoto Prefectural University of Medicine, Kyoto, Japan, 2 Department of Epidemiology for Community Health and Medicine, Graduate School of Medical Sciences, Kyoto Prefectural University of Medicine, Kyoto, Japan, 3 Department of Medical Informatics, Kyoto Prefectural University of Medicine, Kyoto, Japan

☉ These authors contributed equally to this work.
* mami-i78@koto.kpu-m.ac.jp

## Abstract

### Background and objective

The increasing prevalence of atopic dermatitis (AD) has raised concerns about whether individuals with AD require specific cardiovascular disease (CVD) prevention strategies. This study investigated the association between AD and CVD among middle-aged adults.

### Methods

We conducted a nested case–control study using data from the Kyoto Claim Database (April 2013–March 2023) among individuals aged 40–59 years who were followed for ≥ 3 years. Cases were patients with first-onset CVD (hospitalization for ischemic heart disease or stroke), whereas controls had no history of CVD. AD was defined by an ICD-10 code (L20) plus a topical corticosteroid (TCS) prescription. For each case, 10 controls were matched on age, sex, index month, hypertension, diabetes, dyslipidemia, hyperuricemia, and use of anticoagulant or antiplatelet agents. Logistic regression was used to assess associations between CVD and AD prevalence or severity.

### Results

We identified 2,757 CVD cases, including 1,247 with ischemic heart disease and 1,563 with stroke (median age 53 years [interquartile range, 49–56]; 2,031 [73.7%] male). Comorbidities included hypertension in 1,430 (51.9%), diabetes in 583 (21.1%), dyslipidemia in 1,018 (36.9%), hyperuricemia in 307 (11.1%), and anticoagulant or antiplatelet prescriptions in 377 (13.7%). The median follow-up period was

which permits unrestricted use, distribution, and reproduction in any medium, provided the original author and source are credited.

**Data availability statement:** This study used data from the Kyoto Prefecture Claim Database, Japan. The data were provided under restricted access and cannot be shared with third parties because data contain potentially identifying or sensitive patient information. Kyoto Prefecture, Japan, is the owner and custodian of the dataset. contact information: mice@koto.kpu-m.ac.jp.

**Funding:** The author(s) received no specific funding for this work.

**Competing interests:** The authors have declared that no competing interests exist.

60 months. After matching, 2,672 cases and 26,720 controls were compared. AD was diagnosed in 66 cases (2.5%) and 728 controls (2.7%), with no significant association between AD and CVD (odds ratio [OR], 0.90; 95% confidence interval, 0.69–1.16). Regarding AD severity, 3 cases (0.1%) and 76 controls (0.3%) were in the top 10% of average monthly TCS dose (≥37.8 g/month); 28 cases (1.0%) and 352 controls (1.3%) received class 1 TCS; and 14 cases (0.5%) and 144 controls (0.5%) received systemic treatment (immunosuppressants or biologics). AD severity was not associated with CVD risk (ORs: 0.39 [0.10–1.05], 0.79 [0.53–1.15], and 0.97 [0.53–1.62], respectively). A limitation of this study was potential misclassification of AD status due to the nature of claims data.

## Conclusion

Among adults aged 40–59 years, AD was not significantly associated with an increased risk of CVD onset, even in severe cases. Targeted CVD screening for patients with AD may not be necessary; however, comprehensive management of standard CVD risk factors remains essential, as in the general population.

## Introduction

Atopic dermatitis (AD) is a common inflammatory skin disease that affects up to 15%–20% of children and 10% of adults in high-income countries [1,2]. Among skin diseases, AD carries the greatest burden in terms of disability-adjusted life-years (DALYs) [3]. Beyond its cutaneous manifestations, AD is recognized as a systemic inflammatory condition and is associated with multiple comorbidities, including asthma, allergic rhinitis, and food allergy, collectively referred to as the atopic march [4]. Increasing evidence also indicates that AD may contribute to a higher risk of various health conditions, including cardiovascular diseases (CVDs) [5,6]. Given this systemic inflammatory profile, interest has grown in understanding whether AD may contribute to the development of CVD.

Ischemic heart disease (IHD) and stroke remain the leading causes of mortality and DALYs worldwide and in Japan [7,8]. Because inflammation is a major risk factor for CVD [9], the potential link between AD and CVD has gained considerable attention in recent years. However, current evidence is inconsistent regarding whether AD independently increases CVD risk [10–31]. A 2019 systematic review reported a modest association between AD and myocardial infarction or stroke as well as a trend toward increasing CVD risk with greater AD severity; however, substantial heterogeneity across studies limited the certainty of these findings [21]. Another nationwide study found a positive association between AD and CVD compared with the general population; however, this association was attenuated after adjusting for smoking, education, and traditional CVD risk factors [22]. Among adults with AD, lifestyle and metabolic factors, such as obesity [32–34], diabetes mellitus [18,34], hypertension [34,35], dyslipidemia [36,37], smoking [38], alcohol consumption [39], and physical inactivity [39], have been identified as key contributors to CVD risk.

Additionally, one cohort study reported that severe and predominantly active AD may be associated with increased CVD risk [15]. Taken together, the extent to which systemic inflammation associated with AD contributes to CVD development remains uncertain.

To address these gaps, this study aimed to investigate the association between AD (defined by diagnosis and topical corticosteroid (TCS) prescription) and CVD in a middle-aged population, a demographic in which CVD risk begins to increase, using data from a Japanese claims database.

## Methods

### Study design and data source

We conducted a nested case–control study to compare the prevalence of AD between patients with first-onset CVD (cases) and those without CVD (controls) in a population aged 40–59 years. A 3-year observation window prior to the index month was used. Data were obtained from a claims database in Kyoto, Japan (April 1, 2013–March 31, 2023), which covers approximately 60% of the prefecture's residents. The database integrates two major health insurance systems: the National Health Insurance, which covers self-employed individuals, retirees, and their families (database 1), and the Japan Health Insurance Association, which covers employees of small- and medium-sized enterprises and their families (database 2) (Fig 1).

The longitudinal database includes patient characteristics (age and sex), diagnostic information based on International Classification of Diseases, 10th Revision (ICD-10) codes, medical records, hospitalization and outpatient visit data, and death-related information. It provides comprehensive access to medical information from all healthcare institutions visited by insured individuals during their coverage period, even when patients receive care at multiple facilities.

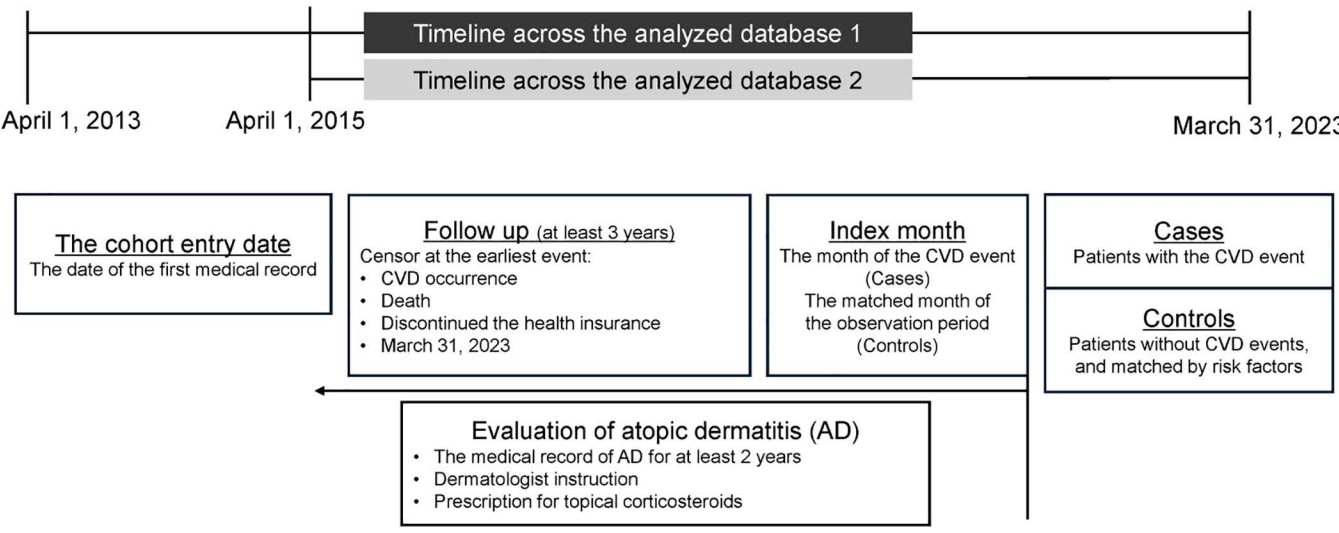

**Fig 1. Overview of the nested case–control study design.** The Kyoto Claim Database consists of health insurance claims from two major insurance systems: the National Health Insurance and the Japan Health Insurance Association. The National Health Insurance (database 1) includes self-employed individuals, retirees, and their families from April 1, 2013, to March 31, 2023. The Japan Health Insurance Association (database 2) covers owners and employees of small- and medium-sized businesses and their families from April 1, 2015, to March 31, 2023. The index month was defined as the month of the first CVD event for cases, or a randomly selected month within the observation period for controls. A 3-year time window preceding the index month was used for exposure assessment. CVD, cardiovascular disease; AD, atopic dermatitis.

## Study participants

Eligible participants were individuals aged 40–59 years at the index month who had been continuously insured for at least 3 years prior to that month. For case patients, the index month was defined as the month of their first CVD event. For control patients, the index month was randomly selected from within the study observation period. Individuals with a history of CVD before the index month were excluded.

## Case identification: CVD

Cases were identified based on a composite CVD outcome, defined as hospitalization for IHD or stroke (cerebral hemorrhage or cerebral infarction), using ICD-10 codes listed in S1 Table. IHD was defined as a new diagnosis of I20, I21, I22, I23, or I24 accompanied by cardiac revascularization procedures, including percutaneous coronary intervention or coronary artery bypass graft surgery. Cerebral hemorrhage and cerebral infarction were defined as new diagnoses of I60–I62 and I63, respectively. Patients with a diagnosis of sequelae of cerebral infarction (I69) were excluded. Based on a previously estimated odds ratio of 1.4, the required case sample size was calculated to be 2,368 individuals.

## Exposure measurement: AD

AD status was assessed for all cases and controls during the 3-year observation period preceding the index month. AD was defined as meeting all of the following criteria:

1) At least two definitive ICD-10 diagnoses of L20 during the insurance period

2) A minimum of 2 years between the first L20 record and the index month

3) At least one dermatologist instruction record

4) At least one prescription for TCS

To evaluate AD severity, we calculated each patient's average monthly TCS dose by dividing the total prescribed TCS amount by the AD follow-up duration.

Severity was then categorized using three indicators:

1) Top 10% of average monthly TCS prescriptions

2) At least one prescription for class 1 (strongest) TCS (clobetasol propionate or diflorasone diacetate)

3) At least one systemic treatment, including immunosuppressive agents (e.g., corticosteroids, calcineurin inhibitors) or biologics (e.g., dupilumab, baricitinib, abrocitinib, upadacitinib)

## Matching factors

Patients with CVD were matched to non-CVD controls at a 1:10 ratio. Matching was conducted at the index month based on eight factors: age, sex, index month, hypertension, diabetes mellitus, dyslipidemia, hyperuricemia, and use of anticoagulant or antiplatelet agents during the observation period. Hypertension, diabetes mellitus, dyslipidemia, and hyperuricemia were identified using ICD-10 codes (S1 Table) in combination with corresponding prescribed medications (S2 Table) recorded in the same month. A history of anticoagulant or antiplatelet use was defined as at least one prescription prior to the index month. Additionally, a sensitivity analysis was performed by incorporating two further matching factors: the duration of insurance coverage and the number of months with recorded healthcare visits from insurance enrollment to the index month.

## Statistical analysis

Patient characteristics were summarized as medians with interquartile ranges (IQRs) for continuous variables and as frequencies for categorical variables. Differences between cases and controls were assessed using the Mann–Whitney U test for continuous variables and the $\chi^2$ test or Fisher's exact test for categorical variables. To evaluate the association between AD and CVD, we employed a 1:10 matched case–control design, selecting 10 non-CVD controls for each CVD case based on age, sex, index month, hypertension, diabetes mellitus, dyslipidemia, hyperuricemia, and use of anticoagulant or antiplatelet agents. Logistic regression models were used to estimate odds ratios (ORs) with 95% confidence intervals (CIs) for AD prevalence and severity in relation to CVD risk. A sensitivity analysis was conducted using the same approach but included two additional matching factors. All statistical tests were two-tailed, with $P$ values <0.05 considered significant. Analyses were performed using R version 4.5.1 (R Foundation for Statistical Computing, Vienna, Austria).

## Ethical considerations

The claims database used in this study was fully anonymized, and researchers did not have access to any personally identifiable information during or after data collection. All analyses were conducted on de-identified data in accordance with the Declaration of Helsinki and Japan's national ethics guidelines; therefore, the requirement for individual informed consent was waived. Data analysis was performed between February 26, 2025, and March 31, 2025. The study protocol was approved by the Institutional Review Board of the Kyoto Prefectural University of Medicine (ERB-C-3411).

# Results

## Study participants

Among 78,196 patients with a first CVD event recorded in the databases (40,864 with IHD and 39,130 with stroke), 2,757 individuals were aged 40–59 years at the index month and had at least 3 years of prior observation (Fig 2). Table 1 summarizes the characteristics of these cases. The composite CVD outcome included 1,247 cases (45.2%) of IHD and 1,563 cases (56.7%) of stroke; 62 patients (2.2%) died during the event month. The median age of case patients was 53 years

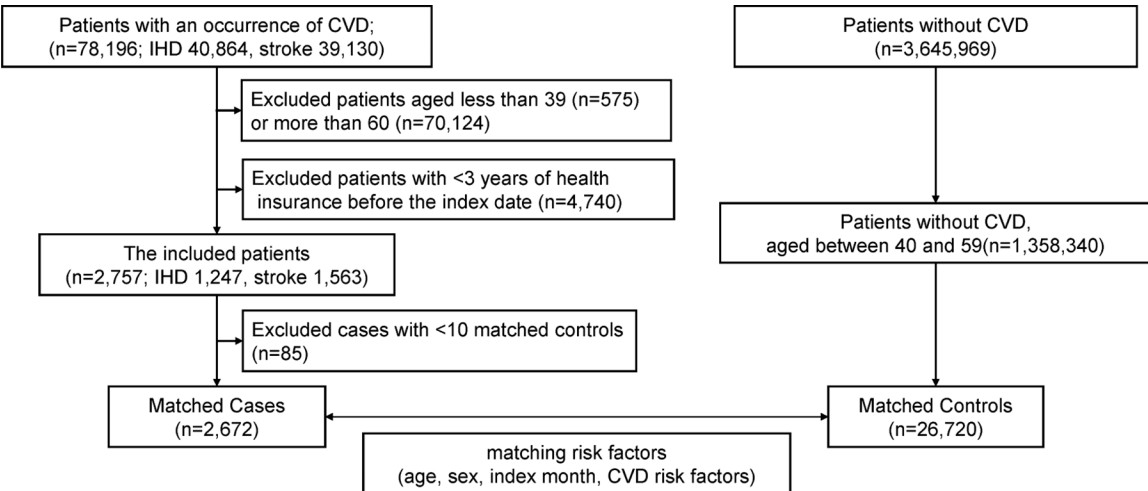

**Fig 2. Patient flow diagram.** Of the 78,196 patients with a first CVD event in the databases (IHD, n = 40,864; stroke, n = 39,130), 2,757 individuals (IHD, n = 1,247; stroke, n = 1,563) were aged 40–59 years at the index month and had at least 3 years of prior observation. Of these, 2,672 were matched with 26,720 controls using eight predefined matching factors for the main analysis. CVD, cardiovascular disease; IHD, ischemic heart disease.

**Table 1. Characteristics of CVD Cases.**

|  | Case, n = 2,757 |
|---|---|
| Age, median (IQR) | 53 [49–56] |
| Sex, male, n (%) | 2031 (73.7) |
| Composite outcome of CVDs |  |
| Ischemic heart disease | 1247 (45.2) |
| Stroke | 1563 (56.7) |
| Death | 62 (2.2) |
| Follow-up duration, median (IQR) | 60 [46–77] |
| Presence of AD, n (%) | 66 (2.4) |
| Presence of severe AD, n (%) |  |
| Prescription of class 1 TCS | 28 (1.0) |
| Systemic treatment | 14 (0.5) |
| Average monthly TCS dose among top 10% (g/month) | 37.8 |
| Hypertension, n (%) | 1430 (51.9) |
| Diabetes mellitus, n (%) | 583 (21.1) |
| Dyslipidemia, n (%) | 1018 (36.9) |
| Hyperuricemia, n (%) | 307 (11.1) |
| Anticoagulant/antiplatelet prescription, n (%) | 377 (13.7) |

Abbreviations: IQR, interquartile range; CVDs, cardiovascular diseases; AD, atopic dermatitis; TCS, topical corticosteroids

(IQR, 49–56), 2,031 (73.7%) were male, and the median follow-up duration was 60 months (IQR, 46–77). Comorbidities included hypertension in 1,430 (51.9%), diabetes mellitus in 583 (21.1%), dyslipidemia in 1,018 (36.9%), and hyperuricemia in 307 (11.1%). Additionally, 377 patients (13.7%) had been prescribed anticoagulants or antiplatelet agents. When comparing patient characteristics by disease type, those with IHD were older, more frequently male, and had a higher prevalence of coronary risk factors than those with stroke (S3 Table). Of the 2,757 eligible cases, 2,672 were successfully matched to 26,720 controls using the eight predefined matching factors for the main analysis. The characteristics of the matched sample closely reflected those of the overall case group (Table 2). Follow-up durations were similar between groups, although the number of practice months differed (Fig 3). For the sensitivity analysis, 2,220 cases were matched with 22,200 controls using ten matching factors. These matched characteristics showed slightly fewer coronary risk factors, consistent with the full case population (S4 Table).

## Main analysis

Among the 2,672 CVD cases and 26,720 matched controls, 66 cases (2.5%) and 728 controls (2.7%) were diagnosed with AD. Regarding AD severity, 3 cases (0.1%) and 76 controls (0.3%) were within the top 10% of average monthly TCS use (45.8 g/month); 28 cases (1.0%) and 325 controls (1.3%) had received class 1 TCS; and 14 cases (0.5%) and 144 controls (0.5%) had received systemic treatment, including oral corticosteroids, calcineurin inhibitors, dupilumab, or baricitinib. There was no significant association between AD and CVD occurrence (OR, 0.90; 95% CI, 0.69–1.16). Similarly, no associations were observed for AD severity indicators: top 10% of average monthly TCS prescriptions (OR, 0.39; 95% CI, 0.10–1.05), class 1 TCS use (OR, 0.79; 95% CI, 0.53–1.15), or systemic treatment (OR, 0.97; 95% CI, 0.53–1.62) (Tables 3 and 4). Cases had slightly lower average monthly TCS prescriptions and fewer AD-related visits than controls. Similar findings were observed in analyses stratified by CVD subtype (IHD or stroke) (S5, S6, S7 and S8 Tables).

**Table 2. Characteristics of cases and matched controls in the main analysis.**

|  | Cases, n = 2,672 | Controls, n = 26,720 |
|---|---|---|
| Matched factors |  |  |
| Age, median (IQR) | 53 [49–57] | 53 [49–56] |
| Sex, male, n (%) | 1976 (74.0) | 19,760 (74.0) |
| Hypertension, n (%) | 1363 (51.0) | 13,630 (51.0) |
| Diabetes mellitus, n (%) | 519 (19.4) | 5190 (19.4) |
| Dyslipidemia, n (%) | 961 (36.0) | 9610 (36.0) |
| Hyperuricemia, n (%) | 255 (9.5) | 2550 (9.5) |
| Anticoagulant/antiplatelet prescription, n (%) | 303 (11.3) | 3030 (11.3) |
| Unmatched factors |  |  |
| Follow-up duration, median (IQR) | 60 [46–78] | 61 [47–77] |
| Number of practice months, median (IQR) | 29 [16–46] | 55 [33–79] |

Abbreviations: IQR, interquartile range.

Matching factors: age, sex, index month, hypertension, diabetes mellitus, dyslipidemia, hyperuricemia, anticoagulant/antiplatelet prescription.

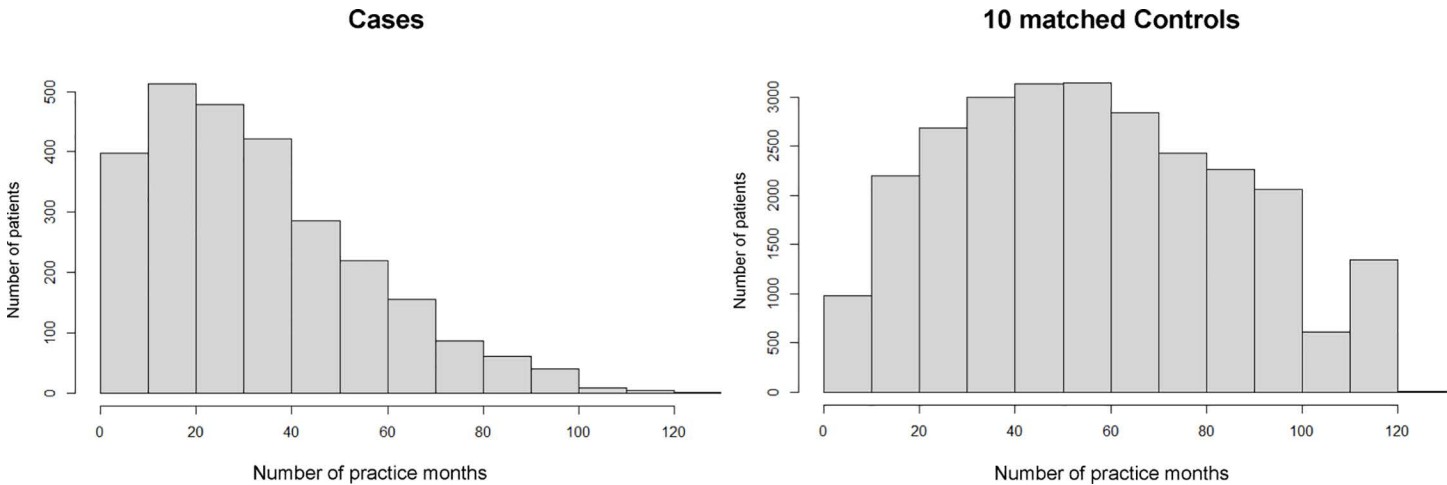

**Fig 3. Distribution of practice months among cases and controls.** The figure illustrates the distribution of the number of practice months during the observation period for cases and controls. Compared with controls, cases tended to have fewer practice months. Median (interquartile range) practice months were 29 (16–46) for cases and 55 (33–79) for controls.

## Sensitivity analysis

A sensitivity analysis was conducted using 10 matching factors, incorporating age (±1 year), duration of insurance coverage (±12 months), number of practice months (±10 months), and exact matching on all other variables. Among the 2,220 cases and 22,200 matched controls, 57 cases (2.6%) and 373 controls (1.7%) were diagnosed with AD. In this analysis, AD prevalence was associated with an increased risk of CVD (OR, 1.54; 95% CI, 1.15–2.02). Partial associations were also observed for AD severity indicators: top 10% of average monthly TCS prescriptions (OR, 1.03; 95% CI, 0.31–2.57), class 1 TCS use (OR, 1.59; 95% CI, 1.01–2.38), and systemic treatments (OR, 1.71; 95% CI, 0.82–3.20) (S7 Table). Similar patterns were observed in analyses stratified by IHD and stroke (S10, S11, S12 and S13 Tables).

**Table 3. Comparison of AD characteristics between cases and matched controls in the main analysis.**

| | Cases, n = 2,672 | Controls, n = 26,720 | OR (95% CIs) |
|---|---|---|---|
| Prevalence of AD, n (%) | 66 (2.5) | 728 (2.7) | 0.90 (0.69–1.16) |
| Prevalence of severe AD, n (%) | | | |
| Prescription for the top 10% of average monthly TCS dose (45.8 g/month) | | | |
| Yes (severe) | 3 (0.1) | 76 (0.3) | 0.39 (0.10–1.05) |
| No (mild) | 63 (2.4) | 652 (2.4) | 0.96 (0.73–1.24) |
| Use of class 1 TCS | | | |
| Yes (severe) | 28 (1.0) | 352 (1.3) | 0.79 (0.53–1.15) |
| No (mild) | 38 (1.4) | 376 (1.4) | 1.01 (0.71–1.39) |
| Systemic treatment | | | |
| Yes (severe) | 14 (0.5) | 144 (0.5) | 0.97 (0.53–1.62) |
| No (mild) | 52 (1.9) | 584 (2.2) | 0.89 (0.66–1.17) |
| Content of systemic treatment | | | |
| Oral corticosteroid | 13 (0.5) | 122 (0.5) | |
| Calcineurin inhibitors | 2 (0.07) | 24 (0.09) | |
| Dupilumab | 0 | 14 (0.05) | |
| Baricitinib | 0 | 1 (0.003) | |

**Table 4. Comparison of AD characteristics between cases and matched controls in the main analysis (Continued).**

| | Cases, n = 2,672 | Controls, n = 26,720 | p value |
|---|---|---|---|
| TCS, monthly average, g, median (IQR) | 8.6 [3.0-16.9] | 10.4 [3.2-26.2] | 0.22 |
| Top 10% for average monthly TCS dose, g | 37.8 | 47.9 | |
| Follow-up duration of AD, median (IQR) | 59.5 [44.2-69.5] | 58 [43-77] | 0.49 |
| Number of practice months of AD, median (IQR) | 15 [8.25-28.3] | 20 [9-37] | 0.15 |

Abbreviation: OR; odds ratio, IQR; interquartile range, AD; atopic dermatitis, TCS; topical corticosteroids.

## Discussion

This study found no significant association between AD and the onset of CVD in a middle-aged population, a group in which CVD risk begins to increase and in which prolonged systemic inflammation from chronic AD might be expected to exert greater influence. Overall, our findings indicate that within this age group, AD (even at higher treatment-defined severity) does not appear to contribute meaningfully to early CVD risk.

The pathophysiology of AD is not fully understood [40], but systemic inflammation has been proposed as a potential contributor to increased CVD risk. Our previous studies indicated that activated platelets may play a role in AD pathogenesis [41–43]. Silverberg et al. reported higher odds of CVD among individuals with AD [12], and a nationwide study found an exposure–response association between AD and ischemic stroke risk [11]. Conversely, several recent meta-analyses have shown only a slight positive association between AD and CVD, particularly among patients with severe AD; however, the findings remain inconsistent due to substantial heterogeneity across studies [18,19,21,31]. This heterogeneity likely reflects wide variation in AD diagnostic criteria, disease severity, and the presence of comorbid conditions driven by systemic inflammation and influenced by complex pathophysiologic mechanisms, genetic background, and diverse environmental factors [24]. Large database studies have also indicated that AD may contribute to CVD risk in adults, but the estimated effect size appears minimal [22,23]. Reflecting this uncertainty, U.S. guidelines note that severe AD in adults may be associated with myocardial infarction, but the evidence is of low certainty, and the relationship with stroke remains unclear [44].

Japanese guidelines do not currently address CVD risk management in patients with AD [45–47]. A large cohort study in Japan using a claims database examined 691,338 individuals with AD and reported a slightly higher incidence of CVD compared with those without AD [48]. However, the study population consisted predominantly of younger individuals (half were <19 years old, and only 16% were aged 40–59 years). Conversely, our study specifically targeted an age group at higher baseline risk for CVD; however, no association was observed between AD and CVD development.

The rationale for adopting a case–control design was twofold [49]. First, although AD-related inflammation may take decades to contribute to atherosclerosis and subsequent CVD, the median follow-up duration in our database was only 5 years. This limited time frame makes it difficult to adequately assess long-term associations using a cohort design. Second, a claims database with a large enrolled population is well suited for a case–control approach. By restricting the study population to individuals with at least 3 years of observation and a minimum 2-year history of AD before the first CVD event, we were able to evaluate CVD occurrence among those with persistent AD-related inflammation. The similar follow-up durations observed between cases and controls indicate that time-window bias was unlikely [50]. Additionally, we used a 1:10 matching strategy, selecting as many suitable controls as possible while matching on all relevant patient characteristics without reducing the sample size [51].

Beyond heterogeneity across studies, several methodological considerations specific to claims-based analyses may influence our findings. For instance, misclassification of AD may contribute to inconsistent findings across previous studies. Data from the Japanese AD registry indicate that most patients with moderate or more severe AD continue to receive TCS prescriptions over a 2-year period, even when clinical severity improves [52,53]. To better identify individuals with ongoing disease activity requiring sustained treatment, we defined AD based on both TCS prescriptions and diagnostic records. Additionally, we required at least one record of "dermatologist instruction," thereby increasing diagnostic accuracy by ensuring that each patient had been evaluated by a dermatologist at least once. To define severe AD, we used three criteria: being in the top 10% of average monthly TCS prescriptions, receiving class 1 TCS, or receiving systemic treatment. Several studies have also classified severe AD based on systemic treatment use [11,13,15,22,27]. A key strength of our study is the incorporation of TCS dosage and potency as severity indicators, as disease activity scores, such as the Eczema Area and Severity Index (EASI), Scoring Atopic Dermatitis (SCORAD), or Patient-Oriented Eczema Measure [54], were not available in the claims database.

Another source of potential misclassification in our study relates to differences in the types of systemic treatment prescribed for severe AD between cases and controls. Among cases, systemic therapy consisted only of oral corticosteroids and/or calcineurin inhibitors, whereas controls also received biologics such as dupilumab or baricitinib. This discrepancy may have introduced misclassification, as biologics could potentially reduce CVD risk by suppressing AD disease activity. Moreover, cases had fewer AD-related visits than controls, which may have led to an underestimation of AD prevalence and severity among cases. To address differences in healthcare utilization and access, we conducted a sensitivity analysis incorporating two additional matching factors: the duration of insurance coverage and the number of any practice months from insurance enrollment to the index month. This analysis showed a positive association between AD and CVD. By adjusting for healthcare utilization, individuals with high TCS use (who were more common in the control group) were excluded. The prevalence of CVD risk factors was also lower in the sensitivity analysis than in the main analysis. These findings indicate that the relationship between AD and CVD onset may depend on both AD severity and the underlying cardiovascular risk profile. To clarify the nature of this association, future studies should employ prospective designs with detailed clinical information and stratified analyses based on AD severity.

This study excluded individuals with a CVD diagnosis who had no history of CVD-related angioplasty, surgery, or hospitalization during the observation period. This criterion may introduce selection bias, as patients with severe AD who received systemic therapies, such as corticosteroids, cyclosporine, or Janus kinase inhibitors, might have developed CVD earlier and thus been excluded. Such therapies could elevate CVD risk, potentially removing high-risk AD patients from the study population. However, when we compared AD prevalence and the proportion of patients receiving systemic

treatment between the excluded group and the study cohort, we found no evidence that the excluded population had more severe AD (S14 Table).

One strength of this study is the large size of the claims database, which covers approximately 60% of Kyoto Prefecture's residents and enabled the use of a robust matched case–control design. Additionally, the database includes complete medical examination and treatment information from all healthcare facilities visited by each patient, even when care was received at multiple institutions. This comprehensive coverage allowed for accurate identification of AD and reliable assessment of disease severity.

Our study has some limitations. First, defining AD using claims data, without access to clinical severity measures such as EASI or SCORAD, may have introduced misclassification, particularly for individuals who did not seek medical care or whose disease severity was underestimated. Second, when using average monthly TCS dose as a marker of AD severity, fluctuations in disease activity may not have been captured, as periods of high activity are diluted when averaged over long treatment durations. This limitation was partially addressed through the use of class 1 TCS prescriptions and systemic therapy as additional severity indicators. Third, because the exact onset of AD was unknown, we could not accurately determine disease duration, which limits the ability to assess the long-term impact of chronic inflammation on CVD development. Fourth, hypertension, diabetes mellitus, dyslipidemia, and hyperuricemia were defined based on diagnosis codes and medication use; matching may therefore have been imperfect, as laboratory values and other measures of disease severity were unavailable. Fifth, important coronary risk factors, including smoking, alcohol consumption, body mass index, and physical activity, were not captured in the claims data. Finally, the study population consisted exclusively of Japanese individuals, which may limit the generalizability of the findings to other populations.

In conclusion, this study found no significant association between AD and CVD among middle-aged Japanese individuals, even in those with high AD severity. Although targeted CVD screening may not be a major management priority for patients with AD, comprehensive control of traditional CVD risk factors remains essential, as it is for the general population.

## Supporting information

**S1 Table. ICD-10 codes.**
(DOCX)

**S2 Table. Drugs used to define the matching factors.**
(DOCX)

**S3 Table. Case characteristics of IHD and stroke.**
(DOCX)

**S4 Table. Characteristics of cases and matched controls in the sensitivity analysis.**
(DOCX)

**S5 Table. Characteristics of cases with IHD and matched controls in the main analysis.**
(DOCX)

**S6 Table. Characteristics of cases with stroke and matched controls in the main analysis.**
(DOCX)

**S7 Table. Comparison of AD characteristics between cases with IHD and matched controls in the main analysis.**
(DOCX)

**S8 Table. Comparison of AD characteristics between cases with stroke and matched controls in the main analysis.**
(DOCX)

**S9 Table. Comparison of AD characteristics between cases and matched controls in the sensitivity analysis.**
(DOCX)

**S10 Table. Characteristics of cases with IHD and matched controls in the sensitivity analysis.**
(DOCX)

**S11 Table. Characteristics of cases with stroke and matched controls in the sensitivity analysis.**
(DOCX)

**S12 Table. Comparison of AD characteristics between cases with IHD and matched controls in the sensitivity analysis.**
(DOCX)

**S13 Table. Comparison of AD characteristics between cases with stroke and matched controls in the sensitivity analysis.**
(DOCX)

**S14 Table. Characteristics of AD among the excluded population.**
(DOCX)

## Acknowledgments

We would like to thank J. Okumura for administrative and technical support.

## Author contributions

**Conceptualization:** Misato Maeno, Mami Ishida, Risa Tamagawa-Mineoka, Naoyuki Takashima, Hiroshi Ikai.

**Data curation:** Mami Ishida, Hiroshi Ikai.

**Formal analysis:** Mami Ishida.

**Investigation:** Misato Maeno, Mami Ishida, Hiroshi Ikai.

**Methodology:** Misato Maeno, Mami Ishida, Hiroshi Ikai.

**Project administration:** Hiroshi Ikai.

**Resources:** Naoyuki Takashima, Hiroshi Ikai.

**Software:** Mami Ishida.

**Supervision:** Naoyuki Takashima, Hiroshi Ikai.

**Validation:** Mami Ishida, Naoyuki Takashima, Hiroshi Ikai.

**Visualization:** Mami Ishida.

**Writing – original draft:** Misato Maeno, Mami Ishida.

**Writing – review & editing:** Misato Maeno, Mami Ishida, Risa Tamagawa-Mineoka, Naoyuki Takashima, Hiroshi Ikai.

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
