## [Decision Letter · Decision Letter 0]

7 Nov 2025

Dear Dr. Ishida,

Thank you for submitting your manuscript to PLOS ONE. After careful consideration, we feel that it has merit but does not fully meet PLOS ONE’s publication criteria as it currently stands. Therefore, we invite you to submit a revised version of the manuscript that addresses the points raised during the review process.

We look forward to receiving your revised manuscript.

Kind regards,

Seyed Aria Nejadghaderi

Academic Editor

PLOS ONE

Journal Requirements:

Reviewers' comments:

Reviewer's Responses to Questions

**Comments to the Author**

1. Is the manuscript technically sound, and do the data support the conclusions?

Reviewer #1: Yes

Reviewer #2: Yes

2. Has the statistical analysis been performed appropriately and rigorously?

Reviewer #1: Yes

Reviewer #2: I Don't Know

3. Have the authors made all data underlying the findings in their manuscript fully available?

Reviewer #1: Yes

Reviewer #2: Yes

4. Is the manuscript presented in an intelligible fashion and written in standard English?

Reviewer #1: Yes

Reviewer #2: Yes

Reviewer #1: This manuscript investigates the association between atopic dermatitis (AD) and cardiovascular disease (CVD) in a middle-aged population in Japan. The authors employed a nested case-control design, utilizing data from the Kyoto Claim Database spanning ten years (April 2013 to March 2023). The study focuses on individuals aged 40–59 years, comparing first-onset CVD cases with matched controls without prior CVD history. The diagnosis of AD was defined based on both ICD-10 codes and prescribed topical corticosteroids, while logistic regression analysis was used to explore associations between AD, its severity, and CVD outcomes. The results indicate that, among the study population, there was no statistically significant association between AD and CVD onset. The authors report that the odds ratios for CVD occurrence did not support a relationship with either the presence or severity of AD. The paper acknowledges its limitations, particularly around potential misclassification of disease prevalence and severity inherent to claim-based data.

comments:

The manuscript's similarity index is high. The authors should work to reduce this significantly to ensure originality and clarity.

The abstract should encapsulate a stronger public health message to highlight the implications of the findings for broader cardiovascular disease prevention strategies.

The introduction would benefit from a clearer transition between the problem statement and the study objectives, helping readers understand the motivation behind the research.

Captions for figures and tables should be improved to be more informative, enhancing reader comprehension of the data presented.

The manuscript lacks details on the clinical settings in which the findings are applicable. Providing this context is crucial for interpreting risk and deriving meaningful conclusions.

The limitations section should be expanded to include:

Underreporting and variable data quality

Confounding variables and the need for adjustment for comorbidities

The inability to establish causality

Absence of data on treatment settings and disease severity

Lack of advanced stratification to explore heterogeneity in the population

Please provide future research direction for the study

I encourage the authors to consider performing external validation of their findings using other databases to enhance the robustness of their conclusions.

The manuscript should clearly state the study sample size calculation, which is essential for understanding the statistical power of the findings.

To improve discoverability, please provide keywords aligned with MeSH (Medical Subject Headings).

The authors should elaborate on the specific steps taken in their analysis framework, detailing how model parameters and their corresponding confidence intervals (CIs) were derived.

The manuscript needs to clarify how confounding factors were controlled in the analysis, aiding readers in understanding the study's integrity.

I suggest that the authors discuss how the findings might apply to other populations or regions with differing contexts, potentially broadening the impact of their research.

Reviewer #2: Dear authors:

This is a well-organized, clearly written case–control study exploring whether atopic dermatitis is associated with cardiovascular disease risk in a large middle-aged Japanese cohort. However, the following areas need improvement:

1. Abstract could be slightly shortened and should state “no significant association” rather than “no positive association”.

2. The manuscript is clearly written in professional English, though minor grammatical corrections are needed (e.g., “systematic treatment” should be “systemic treatment”; “cardiac artery bypass” is better to be written “coronary artery bypass”).

3. The manuscript includes repetitive content across the Abstract, Results, and Discussion; condensing overlapping information and emphasizing interpretation would enhance clarity and flow.

4. The exclusion of participants with prior cardiovascular disease may introduce selection bias, as some patients with severe atopic dermatitis treated with systemic corticosteroids or JAK inhibitors (both known to increase cardiovascular risk) may have already developed CVD earlier and were therefore excluded from the dataset. This selective exclusion removes individuals at the highest risk, biasing the cohort toward healthier survivors and likely underestimating the true association between atopic dermatitis and cardiovascular disease.

5. The matching strategy (age, sex, major comorbidities) is appropriate; however, key behavioral confounders( smoking, BMI, physical activity, alcohol)are missing. This must be discussed more explicitly in the Limitations as residual confounding may mask true associations.

6. Although PLOS ONE allows format-free initial submission, the current placement of tables, figures, and the Sensitivity Analysis makes the manuscript hard to follow. placing figures and tables near their first mention or and the end would improve readability, while final formatting can follow PLOS ONE guidelines after acceptance.

**Do you want your identity to be public for this peer review?** For information about this choice, including consent withdrawal, please see our Privacy Policy

Reviewer #1: No

Reviewer #2: **Yes:** Shirin Zaresharifi

---

## [Author Response · Author response to Decision Letter 1]

24 Dec 2025

Response to Editor

(Response) We confirmed it.

(Response) We confirmed it.

(Response) We updated our Data Availability statement in the submission form accordingly.　This study used data obtained from the Kyoto Prefecture Claims Database, Japan. The data were provided under restricted access and cannot be shared with third parties. Kyoto Prefecture, Japan, is the owner and custodian of the dataset.

(Response) Because we have restrictions on sharing a de-identified data set, we provided this reason in the Data Availability Statement submission form.

(Response) There were no recommendations to add specific previously published works.

Response to Reviewer #1

(Comment 1)

The manuscript's similarity index is high. The authors should work to reduce this significantly to ensure originality and clarity.

(Response)

Thank you for your comments.

We checked the similarity index of the manuscript using iThenticate. As you mentioned, the similarity index was high; however, this was largely due to the frequent use of terminologies common to the subject area in previously published work. Notably, there is an abundance of research papers published in this field. Furthermore, as the Introduction section was thought to rely heavily on citations from review articles, we re-examined whether all sources were appropriately referenced. We assure you that the research design is original, and we have substantially rewritten it to use original terminology as much as possible.

(Comment 2)

The abstract should encapsulate a stronger public health message to highlight the implications of the findings for broader cardiovascular disease prevention strategies.

(Response)

Thank you for your comments, but our findings did not strongly suggest that patients with atopic dermatitis (AD) should undergo more thorough cardiovascular disease (CVD) prevention strategies than those without AD. At the same time, we do not believe that CVD prevention should be neglected in patients with AD; we consider it to be just as necessary as it is for the general population. We added the sentence ‘Targeted CVD screening for patients with AD may not be necessary; however, comprehensive management of standard CVD risk factors remains essential, as in the general population.’ in the conclusion.

(Comment 3)

The introduction would benefit from a clearer transition between the problem statement and the study objectives, helping readers understand the motivation behind the research.

(Response)

We have substantially rewritten the manuscript for clarity and better readability.

(Comment 4)

Captions for figures and tables should be improved to be more informative, enhancing reader comprehension of the data presented.

(Response)

Thank you for your points. We revised the captions for figures and tables.

(Comment 5)

The manuscript lacks details on the clinical settings in which the findings are applicable. Providing this context is crucial for interpreting risk and deriving meaningful conclusions.

(Response)

We described the clinical settings in the ’Study design and Data source’ of the Methods. We revised it for clarity and better readability.

(Comment 6)

The limitations section should be expanded to include:

Underreporting and variable data quality

Confounding variables and the need for adjustment for comorbidities

The inability to establish causality

Absence of data on treatment settings and disease severity

Lack of advanced stratification to explore heterogeneity in the population

(Response)

>Underreporting and variable data quality

If it refers to limitations based on the characteristics of the insurance claims database, then we amended the first sentence of the limitations as follows: ‘First, defining AD using claims data, without access to clinical severity measures such as EASI or SCORAD, may have introduced misclassification, particularly for individuals who did not seek medical care or whose disease severity was underestimated.’

>Confounding variables and the need for adjustment for comorbidities

Confounding factors that could be defined were adjusted for through matching.

Factors that were not defined or obtained could not be adjusted for, as stated in the limitation, ‘Fifth, important coronary risk factors, including smoking, alcohol consumption, body mass index, and physical activity, were not captured in the claims data.’

>The inability to establish causality

As this is an observational study, it is not designed to establish causality; therefore, it is not considered necessary to explicitly state this as a limitation of the present research.

>Absence of data on treatment settings and disease severity

We wrote it as a misclassification in the first limitation. We revised it as follows: ‘First, defining AD using claims data, without access to clinical severity measures such as EASI or SCORAD, may have introduced misclassification, particularly for individuals who did not seek medical care or whose disease severity was underestimated.

>Lack of advanced stratification to explore heterogeneity in the population

We added that ‘Finally, the study population consisted exclusively of Japanese individuals, which may limit the generalizability of the findings to other populations.’

(Comment 7)

Please provide future research direction for the study.

(Response)

Thank you for your comments. We wrote the following: ‘To clarify the nature of this association, future studies should employ prospective designs with detailed clinical information and stratified analyses based on AD severity.’ in the last sentence of the 6th paragraph in the Discussion section.

(Comment 8)

I encourage the authors to consider performing external validation of their findings using other databases to enhance the robustness of their conclusions.

(Response)

Thank you for your suggestion. As this is not feasible within the scope of the current research, we will consider it as a future prospect.

(Comment 9)

The manuscript should clearly state the study sample size calculation, which is essential for understanding the statistical power of the findings.

(Response)

Thank you for your comments. The entire population in this database was included as the study cohort. Having previously estimated the odds ratio of 1.4, the required case sample size was calculated at 2,368 individuals; therefore, the current case participants of approximately 2,700 individuals were deemed sufficient. We added that in the ‘Case identification: CVD’ in the Method section.

(Comment 10)

To improve discoverability, please provide keywords aligned with MeSH (Medical Subject Headings).

(Response)

Thank you for pointing that out. We revised it as follows: “Dermatitis, Atopic, Cardiovascular Diseases, Case-Control Studies, Insurance Claim Review, Middle Aged”.

(Comment 11)

The authors should elaborate on the specific steps taken in their analysis framework, detailing how model parameters and their corresponding confidence intervals (CIs) were derived.

(Response)

We revised ‘Statistical analysis’ clearly as follows: To evaluate the association between AD and CVD, we employed a 1:10 matched case–control design, selecting 10 non-CVD controls for each CVD case based on age, sex, index month, hypertension, diabetes mellitus, dyslipidemia, hyperuricemia, and use of anticoagulant or antiplatelet agents. Logistic regression models were used to estimate odds ratios (ORs) with 95% confidence intervals (CIs) for AD prevalence and severity in relation to CVD risk.’

(Comment 12)

The manuscript needs to clarify how confounding factors were controlled in the analysis, aiding readers in understanding the study's integrity.

(Response)

Thank you for your comments. We revised it clearly in the paragraph of ‘Matching factors’ and ‘Statistical analysis’ in the Method section.

(Comment 13)

I suggest that the authors discuss how the findings might apply to other populations or regions with differing contexts, potentially broadening the impact of their research.

(Response)

We added that ‘Finally, the study population consisted exclusively of Japanese individuals, which may limit the generalizability of the findings to other populations,’ as the last sentence in the Limitations.

RESPONSE TO REVIEWER #2

(Comment 14)

Abstract could be slightly shortened and should state “no significant association” rather than “no positive association”.

(Response)

Thank you for your comments. We revised the abstract.

(Comment 15)

The manuscript is clearly written in professional English, though minor grammatical corrections are needed (e.g., “systematic treatment” should be “systemic treatment”; “cardiac artery bypass” is better to be written “coronary artery bypass”).

(Response)

Thank you for pointing that out. We revised them as you mentioned.

(Comment 16)

The manuscript includes repetitive content across the Abstract, Results, and Discussion; condensing overlapping information and emphasizing interpretation would enhance clarity and flow.

(Response)

Thank you for your comments. We have rewritten it to clear the content.

(Comment 17)

The exclusion of participants with prior cardiovascular disease may introduce selection bias, as some patients with severe atopic dermatitis treated with systemic corticosteroids or JAK inhibitors (both known to increase cardiovascular risk) may have already developed CVD earlier and were therefore excluded from the dataset. This selective exclusion removes individuals at the highest risk, biasing the cohort toward healthier survivors and likely underestimating the true association between atopic dermatitis and cardiovascular disease.

(Response)

Thank you for your comments. As you mentioned, individuals with a diagnosis of CVD who had no episodes of CVD-related angioplasty, surgery, or hospitalization during the observation period were excluded from the study. When comparing the prevalence of AD and the proportion of patients receiving systemic treatment: corticosteroids, cyclosporine, or JAK inhibitors, it was not revealed that the excluded population had severe AD. We added it to the 7th paragraph, shown in S10 Table.

(Comment 18)

The matching strategy (age, sex, major comorbidities) is appropriate; however, key behavioral confounders (smoking, BMI, physical activity, alcohol) are missing. This must be discussed more explicitly in the Limitations as residual confounding may mask true associations.

(Response)

Thank you for your comments. We revised the sentence as follows: ‘Fifth, important coronary risk factors, including smoking, alcohol consumption, body mass index, and physical activity, were not captured in the claims data.‘ in the limitations.

(Comment 19)

Although PLOS ONE allows format-free initial submission, the current placement of tables, figures, and the Sensitivity Analysis makes the manuscript hard to follow. placing figures and tables near their first mention or and the end would improve readability, while final formatting can follow PLOS ONE guidelines after acceptance.

(Response)

Thank you for your advice. We placed them at the end.

---

## [Decision Letter · Decision Letter 1]

1 Jan 2026

Dear Dr. Ishida,

We look forward to receiving your revised manuscript.

Kind regards,

Seyed Aria Nejadghaderi

Academic Editor

PLOS One

Journal Requirements:

Reviewers' comments:

Reviewer's Responses to Questions

**Comments to the Author**

Reviewer #1: (No Response)

Reviewer #2: (No Response)

2. Is the manuscript technically sound, and do the data support the conclusions?

Reviewer #1: Yes

Reviewer #2: Yes

3. Has the statistical analysis been performed appropriately and rigorously?

Reviewer #1: Yes

Reviewer #2: I Don't Know

4. Have the authors made all data underlying the findings in their manuscript fully available?

Reviewer #1: Yes

Reviewer #2: Yes

5. Is the manuscript presented in an intelligible fashion and written in standard English?

Reviewer #1: Yes

Reviewer #2: Yes

Reviewer #1: (No Response)

Reviewer #2: Dear authors

Thank you for carefully addressing all comments; the manuscript has improved substantially and all requested points have been satisfactorily met. As a final minor editorial correction, please ensure consistent statistical terminology by replacing “no positive association” with “no significant association” in the opening sentence of the Discussion and in the Conclusion, to align with the Abstract and the reported statistical results.

**Do you want your identity to be public for this peer review?** For information about this choice, including consent withdrawal, please see our Privacy Policy

Reviewer #1: No

Reviewer #2: **Yes:** Shirin Zaresahrifi

---

## [Author Response · Author response to Decision Letter 2]

5 Jan 2026

January 05, 2026

Seyed Aria Nejadghaderi

Academic Editor

PLOS ONE

Dear Editor,

We have revised our manuscript in accordance with the detailed suggestions you have graciously provided.

Response to Editor

(Response) There were no recommendations to add specific previously published works.

(Response) We have checked that there are no errors in the list of references. In the twelfth line of the second paragraph of the Introduction, the reference to hypertension has been partially amended. (The citation for reference 35 has been removed.)

RESPONSE TO REVIEWER #2

(Comment)

Thank you for carefully addressing all comments; the manuscript has improved substantially and all requested points have been satisfactorily met. As a final minor editorial correction, please ensure consistent statistical terminology by replacing “no positive association” with “no significant association” in the opening sentence of the Discussion and in the Conclusion, to align with the Abstract and the reported statistical results.

(Response)

Thank you for pointing that out. We revised them as you mentioned.

---

## [Editor Report · Decision Letter 2]

6 Jan 2026

Cardiovascular risk among middle-aged Japanese adults with atopic dermatitis: a nested case–control study

PONE-D-25-50839R2

Dear Dr. Ishida,

We’re pleased to inform you that your manuscript has been judged scientifically suitable for publication and will be formally accepted for publication once it meets all outstanding technical requirements.

Kind regards,

Seyed Aria Nejadghaderi

Academic Editor

PLOS One
---

## [Editor Report · Acceptance letter]

PONE-D-25-50839R2

PLOS One

Dear Dr. Ishida,

I'm pleased to inform you that your manuscript has been deemed suitable for publication in PLOS One. Congratulations! Your manuscript is now being handed over to our production team.

Kind regards,

on behalf of

Dr. Seyed Aria Nejadghaderi

Academic Editor

PLOS One